# On the Dimensionality of Word Embedding

**Zi Yin**
Stanford University
s0960974@gmail.com

**Yuanyuan Shen**
Microsoft Corp. & Stanford University
Yuanyuan.Shen@microsoft.com

## Abstract

In this paper, we provide a theoretical understanding of word embedding and its dimensionality. Motivated by the unitary-invariance of word embedding, we propose the Pairwise Inner Product (PIP) loss, a novel metric on the dissimilarity between word embeddings. Using techniques from matrix perturbation theory, we reveal a fundamental bias-variance trade-off in dimensionality selection for word embeddings. This bias-variance trade-off sheds light on many empirical observations which were previously unexplained, for example the existence of an optimal dimensionality. Moreover, new insights and discoveries, like when and how word embeddings are robust to over-fitting, are revealed. By optimizing over the bias-variance trade-off of the PIP loss, we can explicitly answer the open question of dimensionality selection for word embedding.

## 1 Introduction

Word embeddings are very useful and versatile tools, serving as keys to many fundamental problems in numerous NLP research [Turney and Pantel, 2010]. To name a few, word embeddings are widely applied in information retrieval [Salton, 1971, Salton and Buckley, 1988, Sparck Jones, 1972], recommendation systems [Breese et al., 1998, Yin et al., 2017], image description [Frome et al., 2013], relation discovery [Mikolov et al., 2013c] and word level translation [Mikolov et al., 2013b]. Furthermore, numerous important applications are built on top of word embeddings. Some prominent examples are long short-term memory (LSTM) networks [Hochreiter and Schmidhuber, 1997] that are used for language modeling [Bengio et al., 2003], machine translation [Sutskever et al., 2014, Bahdanau et al., 2014], text summarization [Nallapati et al., 2016] and image caption generation [Xu et al., 2015, Vinyals et al., 2015]. Other important applications include named entity recognition [Lample et al., 2016], sentiment analysis [Socher et al., 2013] and so on.

However, the impact of dimensionality on word embedding has not yet been fully understood. As a critical hyper-parameter, the choice of dimensionality for word vectors has huge influence on the performance of a word embedding. First, it directly impacts the quality of word vectors - a word embedding with a small dimensionality is typically not expressive enough to capture all possible word relations, whereas one with a very large dimensionality suffers from over-fitting. Second, the number of parameters for a word embedding or a model that builds on word embeddings (e.g. recurrent neural networks) is usually a linear or quadratic function of dimensionality, which directly affects training time and computational costs. Therefore, large dimensionalities tend to increase model complexity, slow down training speed, and add inferential latency, all of which are constraints that can potentially limit model applicability and deployment [Wu et al., 2016].

Dimensionality selection for embedding is a well-known open problem. In most NLP research, dimensionality is either selected ad hoc or by grid search, either of which can lead to sub-optimal model performances. For example, 300 is perhaps the most commonly used dimensionality in various studies [Mikolov et al., 2013a, Pennington et al., 2014, Bojanowski et al., 2017]. This is possibly due to the influence of the groundbreaking paper, which introduced the skip-gram Word2Vec model and chose a dimensionality of 300 [Mikolov et al., 2013a]. A better empirical approach used by

some researchers is to first train many embeddings of different dimensionalities, evaluate them on a functionality test (like word relatedness or word analogy), and then pick the one with the best empirical performance. However, this method suffers from 1) greatly increased time complexity and computational burden, 2) inability to exhaust all possible dimensionalities and 3) lack of consensus between different functionality tests as their results can differ. Thus, we need a universal criterion that can reflect the relationship between the dimensionality and quality of word embeddings in order to establish a dimensionality selection procedure for embedding methods.

In this regard, we outline a few major contributions of our paper:

1. We introduce the PIP loss, a novel metric on the dissimilarity between word embeddings;

2. We develop a mathematical framework that reveals a fundamental bias-variance trade-off in dimensionality selection. We explain the existence of an optimal dimensionality, a phenomenon commonly observed but lacked explanations;

3. We quantify the robustness of embedding algorithms using the exponent parameter $\alpha$, and establish that many widely used embedding algorithms, including skip-gram and GloVe, are robust to over-fitting;

4. We propose a mathematically rigorous answer to the open problem of dimensionality selection by minimizing the PIP loss. We perform this procedure and cross-validate the results with grid search for LSA, skip-gram Word2Vec and GloVe on an English corpus.

For the rest of the paper, we consider the problem of learning an embedding for a vocabulary of size $n$, which is canonically defined as $\mathcal{V} = \{1, 2, \cdots, n\}$. Specifically, we want to learn a vector representation $v_i \in \mathbb{R}^d$ for each token $i$. The main object is the **embedding matrix** $E \in \mathbb{R}^{n \times d}$, consisting of the stacked vectors $v_i$, where $E_{i,\cdot} = v_i$. All matrix norms in the paper are Frobenius norms unless otherwise stated.

## 2   Preliminaries and Background Knowledge

Our framework is built on the following preliminaries:

1. Word embeddings are unitary-invariant;

2. Most existing word embedding algorithms can be formulated as low rank matrix approximations, either explicitly or implicitly.

### 2.1   Unitary Invariance of Word Embeddings

The unitary-invariance of word embeddings has been discovered in recent research [Hamilton et al., 2016, Artetxe et al., 2016, Smith et al., 2017, Yin, 2018]. It states that two embeddings are essentially identical if one can be obtained from the other by performing a unitary operation, e.g., a rotation. A unitary operation on a vector corresponds to multiplying the vector by a unitary matrix, *i.e.* $v' = vU$, where $U^T U = UU^T = Id$. Note that a unitary transformation preserves the relative geometry of the vectors, and hence defines an *equivalence class* of embeddings. In Section 3, we introduce the Pairwise Inner Product loss, a unitary-invariant metric on embedding similarity.

### 2.2   Word Embeddings from Explicit Matrix Factorization

A wide range of embedding algorithms use explicit matrix factorization, including the popular Latent Semantics Analysis (LSA). In LSA, word embeddings are obtained by truncated SVD of a signal matrix $M$ which is usually based on co-occurrence statistics, for example the Pointwise Mutual Information (PMI) matrix, positive PMI (PPMI) matrix and Shifted PPMI (SPPMI) matrix [Levy and Goldberg, 2014]. Eigen-words [Dhillon et al., 2015] is another example of this type.

Caron [2001], Bullinaria and Levy [2012], Turney [2012], Levy and Goldberg [2014] described a generic approach of obtaining embeddings from matrix factorization. Let $M$ be the signal matrix (e.g. the PMI matrix) and $M = UDV^T$ be its SVD. A $k$-dimensional embedding is obtained by truncating the left singular matrix $U$ at dimension $k$, and multiplying it by a power of the truncated diagonal matrix $D$, i.e. $E = U_{1:k} D_{1:k,1:k}^{\alpha}$ for some $\alpha \in [0, 1]$. Caron [2001], Bullinaria

and Levy [2012] discovered through empirical studies that different $\alpha$ works for different language tasks. In Levy and Goldberg [2014] where the authors explained the connection between skip-gram Word2Vec and matrix factorization, $\alpha$ is set to $0.5$ to enforce symmetry. We discover that $\alpha$ controls the robustness of embeddings against over-fitting, as will be discussed in Section 5.1.

## 2.3 Word Embeddings from Implicit Matrix Factorization

In NLP, two most widely used embedding models are skip-gram Word2Vec [Mikolov et al., 2013c] and GloVe [Pennington et al., 2014]. Although they learn word embeddings by optimizing over some objective functions using stochastic gradient methods, they have both been shown to be implicitly performing matrix factorizations.

**Skip-gram**    Skip-gram Word2Vec maximizes the likelihood of co-occurrence of the center word and context words. The log likelihood is defined as

$$\sum_{i=0}^{n} \sum_{j=i-w, j \neq i}^{i+w} \log(\sigma(v_j^T v_i)), \text{ where } \sigma(x) = \frac{e^x}{1+e^x}$$

Levy and Goldberg [2014] showed that skip-gram Word2Vec's objective is an implicit symmetric factorization of the Pointwise Mutual Information (PMI) matrix:

$$\text{PMI}_{ij} = \log \frac{p(v_i, v_j)}{p(v_i)p(v_j)}$$

Skip-gram is sometimes enhanced with techniques like negative sampling [Mikolov et al., 2013b], where the signal matrix becomes the Shifted PMI matrix [Levy and Goldberg, 2014].

**GloVe**    Levy et al. [2015] pointed out that the objective of GloVe is implicitly a symmetric factorization of the log-count matrix. The factorization is sometimes augmented with bias vectors and the log-count matrix is sometimes raised to an exponent $\gamma \in [0, 1]$ [Pennington et al., 2014].

## 3    PIP Loss: a Novel Unitary-invariant Loss Function for Embeddings

How do we know whether a trained word embedding is good enough? Questions of this kind cannot be answered without a properly defined loss function. For example, in statistical estimation (*e.g.* linear regression), the quality of an estimator $\hat{\theta}$ can often be measured using the $l_2$ loss $\mathbb{E}[\|\hat{\theta} - \theta^*\|_2^2]$ where $\theta^*$ is the unobserved ground-truth parameter. Similarly, for word embedding, a proper metric is needed in order to evaluate the quality of a trained embedding.

As discussed in Section 2.1, a reasonable loss function between embeddings should respect the unitary-invariance. This rules out choices like direct comparisons, for example using $\|E_1 - E_2\|$ as the loss function. We propose the Pairwise Inner Product (PIP) loss, which naturally arises from the unitary-invariance, as the dissimilarity metric between two word embeddings:

**Definition 1** (PIP matrix). Given an embedding matrix $E \in \mathbb{R}^{n \times d}$, define its associated Pairwise Inner Product (PIP) matrix to be
$$\text{PIP}(E) = EE^T$$

It can be seen that the $(i, j)$-th entry of the PIP matrix corresponds to the inner product between the embeddings for word $i$ and word $j$, i.e. $\text{PIP}_{i,j} = \langle v_i, v_j \rangle$. To compare $E_1$ and $E_2$, two embedding matrices on a common vocabulary, we propose the **PIP loss**:

**Definition 2** (PIP loss). The PIP loss between $E_1$ and $E_2$ is defined as the norm of the difference between their PIP matrices

$$\|\text{PIP}(E_1) - \text{PIP}(E_2)\| = \|E_1 E_1^T - E_2 E_2^T\| = \sqrt{\sum_{i,j}(\langle v_i^{(1)}, v_j^{(1)} \rangle - \langle v_i^{(2)}, v_j^{(2)} \rangle)^2}$$

Note that the $i$-th row of the PIP matrix, $v_i E^T = (\langle v_i, v_1 \rangle, \cdots, \langle v_i, v_n \rangle)$, can be viewed as the relative position of $v_i$ anchored against all other vectors $\{v_1, \cdots, v_n\}$. In essence, the PIP loss measures the vectors' *relative position shifts* between $E_1$ and $E_2$, thereby removing their dependencies on any specific coordinate system. The PIP loss respects the unitary-invariance. Specifically, if $E_2 = E_1 U$

where $U$ is a unitary matrix, then the PIP loss between $E_1$ and $E_2$ is zero because $E_2 E_2^T = E_1 E_1^T$. In addition, the PIP loss serves as a metric of *functionality* dissimilarity. A practitioner may only care about the usability of word embeddings, for example, using them to solve analogy and relatedness tasks [Schnabel et al., 2015, Baroni et al., 2014], which are the two most important properties of word embeddings. Since both properties are tightly related to vector inner products, a small PIP loss between $E_1$ and $E_2$ leads to a small difference in $E_1$ and $E_2$'s relatedness and analogy as the PIP loss measures the difference in inner products[1]. As a result, from both theoretical and practical standpoints, the PIP loss is a suitable loss function for embeddings. Furthermore, we show in Section 4 that this formulation opens up a new angle to understanding the effect of embedding dimensionality with matrix perturbation theory.

## 4 How Does Dimensionality Affect the Quality of Embedding?

With the PIP loss, we can now study the quality of trained word embeddings for any algorithm that uses matrix factorization. Suppose a $d$-dimensional embedding is derived from a signal matrix $M$ with the form $f_{\alpha,d}(M) \triangleq U_{\cdot,1:d} D_{1:d,1:d}^\alpha$, where $M = UDV^T$ is the SVD. In the ideal scenario, a genie reveals a clean signal matrix $M$ (*e.g.* PMI matrix) to the algorithm, which yields the **oracle embedding** $E = f_{\alpha,d}(M)$. However, in practice, there is no magical oil lamp, and we have to estimate $\tilde{M}$ (*e.g.* empirical PMI matrix) from the training data, where $\tilde{M} = M + Z$ is perturbed by the estimation noise $Z$. The **trained embedding** $\hat{E} = f_{\alpha,k}(\tilde{M})$ is computed by factorizing this noisy matrix. To ensure $\hat{E}$ is close to $E$, we want the PIP loss $\|EE^T - \hat{E}\hat{E}^T\|$ to be small. In particular, this PIP loss is affected by $k$, the dimensionality we select for the trained embedding.

Arora [2016] discussed in an article about a mysterious empirical observation of word embeddings: "... *A striking finding in empirical work on word embeddings is that there is a sweet spot for the dimensionality of word vectors: neither too small, nor too large*"[2]. He proceeded by discussing two possible explanations: low dimensional projection (like the Johnson-Lindenstrauss Lemma) and the standard generalization theory (like the VC dimension), and pointed out why neither is sufficient for explaining this phenomenon. While some may argue that this is caused by underfitting/overfitting, the concept itself is too broad to provide any useful insight. We show that this phenomenon can be explicitly explained by a bias-variance trade-off in Section 4.1, 4.2 and 4.3. Equipped with the PIP loss, we give a mathematical presentation of the bias-variance trade-off using matrix perturbation theory. We first introduce a classical result in Lemma 1. The proof is deferred to the appendix, which can also be found in Stewart and Sun [1990].

**Lemma 1.** Let $X, Y$ be two orthogonal matrices of $\mathbb{R}^{n \times n}$. Let $X = [X_0, X_1]$ and $Y = [Y_0, Y_1]$ be the first $k$ columns of $X$ and $Y$ respectively, namely $X_0, Y_0 \in \mathbb{R}^{n \times k}$ and $k \leq n$. Then

$$\|X_0 X_0^T - Y_0 Y_0^T\| = c\|X_0^T Y_1\|$$

where $c$ is a constant depending on the norm only. $c = 1$ for 2-norm and $\sqrt{2}$ for Frobenius norm.

As pointed out by several papers [Caron, 2001, Bullinaria and Levy, 2012, Turney, 2012, Levy and Goldberg, 2014], embedding algorithms can be generically characterized as $E = U_{1:k,\cdot} D_{1:k,1:k}^\alpha$ for some $\alpha \in [0, 1]$. For illustration purposes, we first consider a special case where $\alpha = 0$.

### 4.1 The Bias Variance Trade-off for a Special Case: $\alpha = 0$

The following theorem shows how the PIP loss can be naturally decomposed into a bias term and a variance term when $\alpha = 0$:

**Theorem 1.** Let $E \in \mathbb{R}^{n \times d}$ and $\hat{E} \in \mathbb{R}^{n \times k}$ be the oracle and trained embeddings, where $k \leq d$. Assume both have orthonormal columns. Then the PIP loss has a bias-variance decomposition

$$\|\text{PIP}(E) - \text{PIP}(\hat{E})\|^2 = d - k + 2\|\hat{E}^T E^\perp\|^2$$

*Proof.* The proof utilizes techniques from matrix perturbation theory. To simplify notations, denote $X_0 = E, Y_0 = \hat{E}$, and let $X = [X_0, X_1], Y = [Y_0, Y_1]$ be the complete $n$ by $n$ orthogonal matrices.

Since $k \leq d$, we can further split $X_0$ into $X_{0,1}$ and $X_{0,2}$, where the former has $k$ columns and the latter $d - k$. Now, the PIP loss equals

$$
\begin{aligned}
\|EE^T - \hat{E}\hat{E}^T\|^2 &= \|X_{0,1}X_{0,1}^T - Y_0Y_0^T + X_{0,2}X_{0,2}^T\|^2 \\
&= \|X_{0,1}X_{0,1}^T - Y_0Y_0^T\|^2 + \|X_{0,2}X_{0,2}^T\|^2 + 2\langle X_{0,1}X_{0,1}^T - Y_0Y_0^T, X_{0,2}X_{0,2}^T\rangle \\
&\overset{(a)}{=} 2\|Y_0^T[X_{0,2}, X_1]\|^2 + d - k - 2\langle Y_0Y_0^T, X_{0,2}X_{0,2}^T\rangle \\
&= 2\|Y_0^T X_{0,2}\|^2 + 2\|Y_0^T X_1\|^2 + d - k - 2\langle Y_0Y_0^T, X_{0,2}X_{0,2}^T\rangle \\
&= d - k + 2\|Y_0^T X_1\|^2 = d - k + 2\|\hat{E}^T E^\perp\|^2
\end{aligned}
$$

where in equality (a) we used Lemma 1. $\qquad\square$

The observation is that the right-hand side now consists of two parts, which we identify as bias and variance. The first part $d - k$ is the amount of lost signal, which is caused by discarding the rest $d - k$ dimensions when selecting $k \leq d$. However, $\|\hat{E}^T E^\perp\|$ increases as $k$ increases, as the noise perturbs the subspace spanned by $E$, and the singular vectors corresponding to smaller singular values are more prone to such perturbation. As a result, the optimal dimensionality $k^*$ which minimizes the PIP loss lies in between 0 and $d$, the rank of the matrix $M$.

## 4.2 The Bias Variance Trade-off for the Generic Case: $\alpha \in (0, 1]$

In this generic case, the columns of $E$, $\hat{E}$ are no longer orthonormal, which does not satisfy the assumptions in matrix perturbation theory. We develop a novel technique where Lemma 1 is applied in a telescoping fashion. The proof of the theorem is deferred to the appendix.

**Theorem 2.** Let $M = UDV^T$, $\tilde{M} = \tilde{U}\tilde{D}\tilde{V}^T$ be the SVDs of the clean and estimated signal matrices. Suppose $E = U_{\cdot,1:d}D_{1:d,1:d}^\alpha$ is the oracle embedding, and $\hat{E} = \tilde{U}_{\cdot,1:k}\tilde{D}_{1:k,1:k}^\alpha$ is the trained embedding, for some $k \leq d$. Let $D = diag(\lambda_i)$ and $\tilde{D} = diag(\tilde{\lambda}_i)$, then

$$
\|\text{PIP}(E) - \text{PIP}(\hat{E})\| \leq \sqrt{\sum_{i=k+1}^{d} \lambda_i^{4\alpha}} + \sqrt{\sum_{i=1}^{k}(\lambda_i^{2\alpha} - \tilde{\lambda}_i^{2\alpha})^2} + \sqrt{2}\sum_{i=1}^{k}(\lambda_i^{2\alpha} - \lambda_{i+1}^{2\alpha})\|\tilde{U}_{\cdot,1:i}^T U_{\cdot,i:n}\|
$$

As before, the three terms in Theorem 2 can be characterized into bias and variances. The first term is the bias as we lose part of the signal by choosing $k \leq d$. Notice that the embedding matrix $E$ consists of signal directions (given by $U$) and their magnitudes (given by $D^\alpha$). The second term is the variance on the *magnitudes*, and the third term is the variance on the *directions*.

## 4.3 The Bias-Variance Trade-off Captures the Signal-to-Noise Ratio

We now present the main theorem, which shows that the bias-variance trade-off reflects the "signal-to-noise ratio" in dimensionality selection.

**Theorem 3** (Main theorem). Suppose $\tilde{M} = M + Z$, where $M$ is the signal matrix, symmetric with spectrum $\{\lambda_i\}_{i=1}^{d}$. $Z$ is the estimation noise, symmetric with iid, zero mean, variance $\sigma^2$ entries. For any $0 \leq \alpha \leq 1$ and $k \leq d$, let the oracle and trained embeddings be

$$
E = U_{\cdot,1:d}D_{1:d,1:d}^\alpha, \ \hat{E} = \tilde{U}_{\cdot,1:k}\tilde{D}_{1:k,1:k}^\alpha
$$

where $M = UDV^T$, $\tilde{M} = \tilde{U}\tilde{D}\tilde{V}^T$ are the SVDs of the clean and estimated signal matrices. Then

1. When $\alpha = 0$,

$$
\mathbb{E}[\|EE^T - \hat{E}\hat{E}^T\|] \leq \sqrt{d - k + 2\sigma^2 \sum_{r \leq k, s > d}(\lambda_r - \lambda_s)^{-2}}
$$

2. When $0 < \alpha \leq 1$,

$$
\mathbb{E}[\|EE^T - \hat{E}\hat{E}^T\|] \leq \sqrt{\sum_{i=k+1}^{d} \lambda_i^{4\alpha}} + 2\sqrt{2n}\alpha\sigma\sqrt{\sum_{i=1}^{k}\lambda_i^{4\alpha-2}} + \sqrt{2}\sum_{i=1}^{k}(\lambda_i^{2\alpha} - \lambda_{i+1}^{2\alpha})\sigma\sqrt{\sum_{r \leq i < s}(\lambda_r - \lambda_s)^{-2}}
$$

*Proof.* We sketch the proof for part 2, as the proof of part 1 is simpler and can be done with the same arguments. We start by taking expectation on both sides of Theorem 2:

$$\mathbb{E}[\|EE^T - \hat{E}\hat{E}^T\|] \leq \sqrt{\sum_{i=k+1}^{d} \lambda_i^{4\alpha}} + \mathbb{E}\sqrt{\sum_{i=1}^{k}(\lambda_i^{2\alpha} - \tilde{\lambda}_i^{2\alpha})^2} + \sqrt{2}\sum_{i=1}^{k}(\lambda_i^{2\alpha} - \lambda_{i+1}^{2\alpha})\mathbb{E}[\|\tilde{U}_{\cdot,1:i}^T U_{\cdot,i:n}\|],$$

The first term involves only the spectrum, which is the same after taking expectation. The second term is upper bounded using Lemma 2 below, derived from Weyl's theorem. We state the lemma, and leave the proof to the appendix.

**Lemma 2.** Under the conditions of Theorem 3,

$$\mathbb{E}\sqrt{\sum_{i=1}^{k}(\lambda_i^{2\alpha} - \tilde{\lambda}_i^{2\alpha})^2} \leq 2\sqrt{2n}\alpha\sigma\sqrt{\sum_{i=1}^{k}\lambda_i^{4\alpha-2}}$$

For the last term, we use the Sylvester operator technique by Stewart and Sun [1990]. Our result is presented in Lemma 3, and the proof of which is discussed in the appendix.

**Lemma 3.** For two matrices $M$ and $\tilde{M} = M + Z$, denote their SVDs as $M = UDV^T$ and $\tilde{M} = \tilde{U}\tilde{D}\tilde{V}^T$. Write the left singular matrices in block form as $U = [U_0, U_1]$, $\tilde{U} = [\tilde{U}_0, \tilde{U}_1]$, and similarly partition $D$ into diagonal blocks $D_0$ and $D_1$. If the spectrum of $D_0$ and $D_1$ has separation

$$\delta_k \triangleq \min_{1\leq i\leq k, k<j\leq n}\{\lambda_i - \lambda_j\} = \lambda_k - \lambda_{k+1} > 0,$$

and $Z$ has iid, zero mean entries with variance $\sigma^2$, then

$$\mathbb{E}[\|\tilde{U}_1^T U_0\|] \leq \sigma\sqrt{\sum_{1\leq i\leq k<j\leq n}(\lambda_i - \lambda_j)^{-2}}$$

Now, collect results in Lemma 2 and Lemma 3, we obtain an upper bound approximation for the PIP loss:

$$\mathbb{E}[\|EE^T - \hat{E}\hat{E}^T\|] \leq \sqrt{\sum_{i=k+1}^{d}\lambda_i^{4\alpha}} + 2\sqrt{2n}\alpha\sigma\sqrt{\sum_{i=1}^{k}\lambda_i^{4\alpha-2}} + \sqrt{2}\sum_{i=0}^{k}(\lambda_i^{2\alpha} - \lambda_{i+1}^{2\alpha})\sigma\sqrt{\sum_{r\leq i<s}(\lambda_r - \lambda_s)^{-2}}$$

which completes the proof. $\square$

Theorem 3 shows that when dimensionality is too small, too much signal power (specifically, the spectrum of the signal $M$) is discarded, causing the first term to be too large (high bias). On the other hand, when dimensionality is too large, too much noise is included, causing the second and third terms to be too large (high variance). This explicitly answers the question of Arora [2016].

## 5 Two New Discoveries

In this section, we introduce two more discoveries regarding the fundamentals of word embedding. The first is the relationship between the robustness of embedding and the exponent parameter $\alpha$, with a corollary that both skip-gram and GloVe are robust to over-fitting. The second is a dimensionality selection method by explicitly minimizing the PIP loss between the oracle and trained embeddings[3]. All our experiments use the Text8 corpus [Mahoney, 2011], a standard benchmark corpus used for various natural language tasks.

### 5.1 Word Embeddings' Robustness to Over-Fitting Increases with Respect to $\alpha$

Theorem 3 provides a good indicator for the sensitivity of the PIP loss with respect to over-parametrization. Vu [2011] showed that the approximations obtained by matrix perturbation theory are minimax tight. As $k$ increases, the bias term $\sqrt{\sum_{i=k}^{d}\lambda_i^{4\alpha}}$ decreases, which can be viewed as a *zeroth-order* term because the arithmetic means of singular values are dominated by the large ones.

As a result, when $k$ is already large (say, the singular values retained contain more than half of the total energy of the spectrum), increasing $k$ has only marginal effect on the PIP loss.

On the other hand, the variance terms demonstrate a *first-order* effect, which contains the difference of the singular values, or singular gaps. Both variance terms grow at the rate of $\lambda_k^{2\alpha-1}$ with respect to the dimensionality $k$ (the analysis is left to the appendix). For small $\lambda_k$ (i.e. $\lambda_k < 1$), the rate $\lambda_k^{2\alpha-1}$ increases as $\alpha$ decreases: when $\alpha < 0.5$, this rate can be very large; When $0.5 \leq \alpha \leq 1$, the rate is bounded and sub-linear, in which case the PIP loss will be robust to over-parametrization. In other words, as $\alpha$ becomes larger, the embedding algorithm becomes less sensitive to over-fitting caused by the selection of an excessively large dimensionality $k$. To illustrate this point, we compute the PIP loss of word embeddings (approximated by Theorem 3) for the PPMI LSA algorithm, and plot them for different $\alpha$'s in Figure 1a.

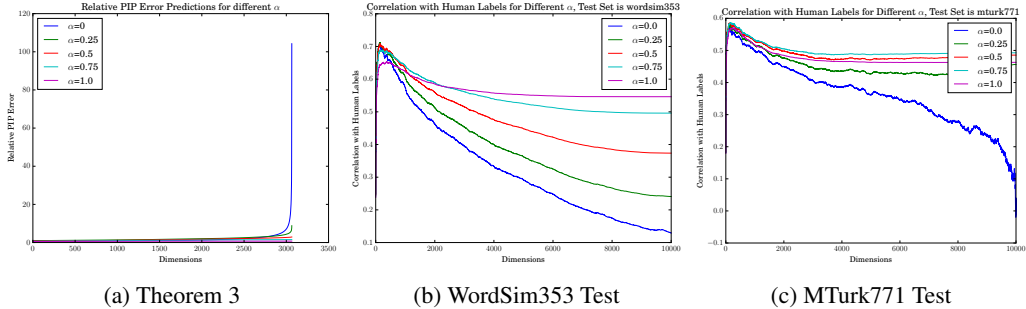

| (a) Theorem 3 | (b) WordSim353 Test | (c) MTurk771 Test |

Figure 1: Sensitivity to over-parametrization: theoretical prediction versus empirical results

Our discussion that over-fitting hurts algorithms with smaller $\alpha$ more can be empirically verified. Figure 1b and 1c display the performances (measured by the correlation between vector cosine similarity and human labels) of word embeddings of various dimensionalities from the PPMI LSA algorithm, evaluated on two word correlation tests: WordSim353 [Finkelstein et al., 2001] and MTurk771 [Halawi et al., 2012]. These results validate our theory: performance drop due to over-parametrization is more significant for smaller $\alpha$.

For the popular skip-gram [Mikolov et al., 2013b] and GloVe [Pennington et al., 2014], $\alpha$ equals 0.5 as they are implicitly doing a symmetric factorization. Our previous discussion suggests that they are robust to over-parametrization. We empirically verify this by training skip-gram and GloVe embeddings. Figure 2 shows the empirical performance on three word functionality tests. Even with extreme over-parametrization (up to $k = 10000$), skip-gram still performs within 80% to 90% of optimal performance, for both analogy test [Mikolov et al., 2013a] and relatedness tests (WordSim353 [Finkelstein et al., 2001] and MTurk771 [Halawi et al., 2012]). This observation holds for GloVe as well as shown in Figure 3.

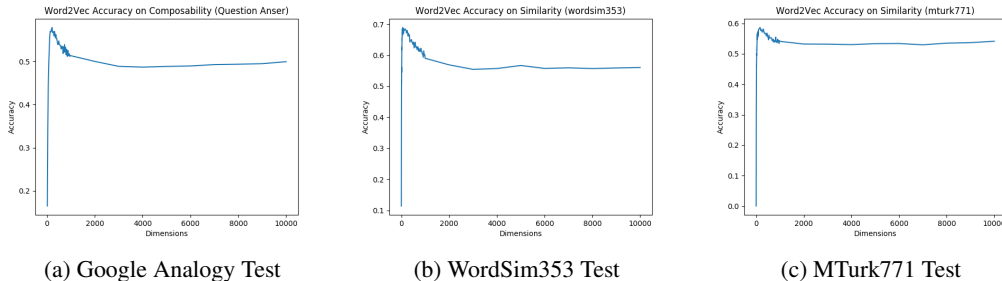

| (a) Google Analogy Test | (b) WordSim353 Test | (c) MTurk771 Test |

Figure 2: skip-gram Word2Vec: over-parametrization does not significantly hurt performance

## 5.2 Optimal Dimensionality Selection: Minimizing the PIP Loss

The optimal dimensionality can be selected by finding the $k^*$ that minimizes the PIP loss between the trained embedding and the oracle embedding. With a proper estimate of the spectrum $D = \{\lambda\}_1^d$

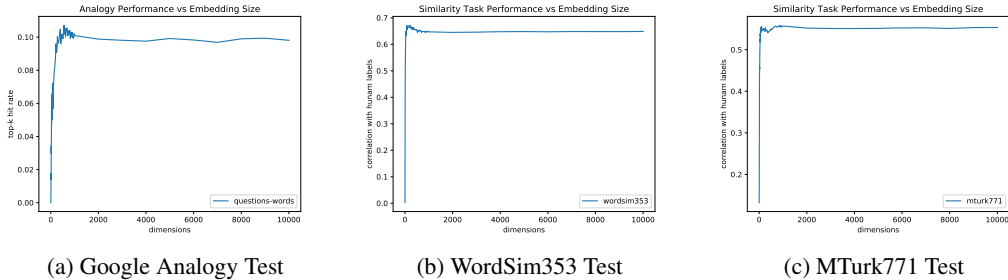

| (a) Google Analogy Test | (b) WordSim353 Test | (c) MTurk771 Test |

Figure 3: GloVe: over-parametrization does not significantly hurt performance

and the variance of noise $\sigma^2$, we can use the approximation in Theorem 3. Another approach is to use the Monte-Carlo method where we simulate the clean signal matrix $M = UDV$ and the noisy signal matrix $\tilde{M} = M + Z$. By factorizing $M$ and $\tilde{M}$, we can simulate the oracle embedding $E = UD^\alpha$ and trained embeddings $\hat{E}_k = \tilde{U}_{\cdot,1:k}\tilde{D}_{1:k,1:k}^\alpha$, in which case the PIP loss between them can be directly calculated. We found empirically that the Monte-Carlo procedure is more accurate as the simulated PIP losses concentrate tightly around their means across different runs. In the following experiments, we demonstrate that dimensionalities selected using the Monte-Carlo approach achieve near-optimal performances on various word intrinsic tests. As a first step, we demonstrate how one can obtain good estimates of $\{\lambda_i\}_1^d$ and $\sigma$ in 5.2.1.

### 5.2.1 Spectrum and Noise Estimation from Corpus

**Noise Estimation** We note that for most NLP tasks, the signal matrices are estimated by counting or transformations of counting, including taking log or normalization. This holds for word embeddings that are based on co-occurrence statistics, *e.g.*, LSA, skip-gram and GloVe. We use a count-twice trick to estimate the noise: we randomly split the data into two equally large subsets, and get matrices $\tilde{M}_1 = M + Z_1$, $\tilde{M}_2 = M + Z_2$ in $\mathbb{R}^{m \times n}$, where $Z_1, Z_2$ are two independent copies of noise with variance $2\sigma^2$. Now, $\tilde{M}_1 - \tilde{M}_2 = Z_1 - Z_2$ is a random matrix with zero mean and variance $4\sigma^2$. Our estimator is the sample standard deviation, a consistent estimator:

$$\hat{\sigma} = \frac{1}{2\sqrt{mn}}\|\tilde{M}_1 - \tilde{M}_2\|$$

**Spectral Estimation** Spectral estimation is a well-studied subject in statistical literature [Cai et al., 2010, Candès and Recht, 2009, Kong and Valiant, 2017]. For our experiments, we use the well-established universal singular value thresholding (USVT) proposed by Chatterjee [2015].

$$\hat{\lambda}_i = (\tilde{\lambda}_i - 2\sigma\sqrt{n})_+$$

where $\tilde{\lambda}_i$ is the $i$-th empirical singular value and $\sigma$ is the noise standard deviation. This estimator is shown to be minimax optimal [Chatterjee, 2015].

### 5.2.2 Dimensionality Selection: LSA, Skip-gram Word2Vec and GloVe

After estimating the spectrum $\{\lambda_i\}_1^d$ and the noise $\sigma$, we can use the Monte-Carlo procedure described above to estimate the PIP loss. For three popular embedding algorithms: LSA, skip-gram Word2Vec and GloVe, we find their optimal dimensionalities $k^*$ that minimize their respective PIP loss. We define the sub-optimality of a particular dimensionality $k$ as the additional PIP loss compared with $k^*$: $\|E_k E_k^T - EE^T\| - \|E_{k^*}E_{k^*}^T - EE^T\|$. In addition, we define the $p\%$ *sub-optimal interval* as the interval of dimensionalities whose sub-optimality are no more than $p\%$ of that of a 1-D embedding. In other words, if $k$ is within the $p\%$ interval, then the PIP loss of a $k$-dimensional embedding is at most $p\%$ worse than the optimal embedding. We show an example in Figure 4.

**LSA with PPMI Matrix** For the LSA algorithm, the optimal dimensionalities and sub-optimal intervals around them (5%, 10%, 20% and 50%) for different $\alpha$ values are shown in Table 1. Figure 4 shows how PIP losses vary across different dimensionalities. From the shapes of the curves, we can see that models with larger $\alpha$ suffer less from over-parametrization, as predicted in Section 5.1.

We further cross-validated our theoretical results with intrinsic functionality tests on word relatedness. The empirically optimal dimensionalities that achieve highest correlations with human labels for the two word relatedness tests (WordSim353 and MTurk777) lie close to the theoretically selected $k^*$'s. All of them fall in the 5% interval except when $\alpha = 0$, in which case they fall in the 20% sub-optimal interval.

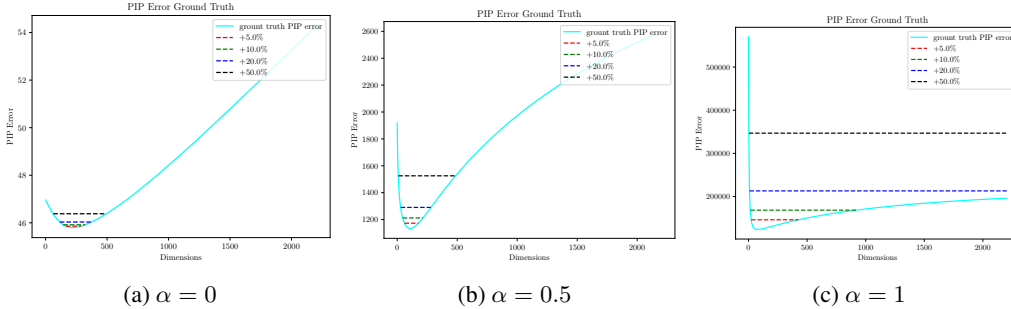

(a) $\alpha = 0$      (b) $\alpha = 0.5$      (c) $\alpha = 1$

Figure 4: PIP loss and its bias-variance trade-off allow for explicit dimensionality selection for LSA

Table 1: Optimal dimensionalities for word relatedness tests are close to PIP loss minimizing ones

| $\alpha$ | PIP arg min | 5% interval | 10% interval | 20% interval | 50% interval | WS353 opt. | MT771 opt. |
|---|---|---|---|---|---|---|---|
| 0 | 214 | [164,289] | [143,322] | [115,347] | [62,494] | 127 | 116 |
| 0.25 | 138 | [95,190] | [78,214] | [57,254] | [23,352] | 146 | 116 |
| 0.5 | 108 | [61,177] | [45,214] | [29,280] | [9,486] | 146 | 116 |
| 0.75 | 90 | [39,206] | [27,290] | [16,485] | [5,1544] | 155 | 176 |
| 1 | 82 | [23,426] | [16,918] | [9,2204] | [3,2204] | 365 | 282 |

**Word2Vec with Skip-gram** For skip-gram, we use the PMI matrix as its signal matrix [Levy and Goldberg, 2014]. On the theoretical side, the PIP loss-minimizing dimensionality $k^*$ and the sub-optimal intervals (5%, 10%, 20% and 50%) are reported in Table 2. On the empirical side, the optimal dimensionalities for WordSim353, MTurk771 and Google analogy tests are 56, 102 and 220 respectively for skip-gram. They agree with the theoretical selections: one is within the 5% interval and the other two are within the 10% interval.

Table 2: PIP loss minimizing dimensionalities and intervals for Skip-gram on Text8 corpus

| Surrogate Matrix | arg min | +5% interval | +10% interval | +20% interval | +50% interval | WS353 | MT771 | Analogy |
|---|---|---|---|---|---|---|---|---|
| Skip-gram (PMI) | 129 | [67,218] | [48,269] | [29,365] | [9,679] | 56 | 102 | 220 |

**GloVe** For GloVe, we use the log-count matrix as its signal matrix [Pennington et al., 2014]. On the theoretical side, the PIP loss-minimizing dimensionality $k^*$ and sub-optimal intervals (5%, 10%, 20% and 50%) are reported in Table 3. On the empirical side, the optimal dimensionalities for WordSim353, MTurk771 and Google analogy tests are 220, 860, and 560. Again, they agree with the theoretical selections: two are within the 5% interval and the other is within the 10% interval.

Table 3: PIP loss minimizing dimensionalities and intervals for GloVe on Text8 corpus

| Surrogate Matrix | arg min | +5% interval | +10% interval | +20% interval | +50% interval | WS353 | MT771 | Analogy |
|---|---|---|---|---|---|---|---|---|
| GloVe (log-count) | 719 | [290,1286] | [160,1663] | [55,2426] | [5,2426] | 220 | 860 | 560 |

The above three experiments show that our method is a powerful tool in practice: the dimensionalities selected according to empirical grid search agree with the PIP-loss minimizing criterion, which can be done simply by knowing the spectrum and noise standard deviation.

# 6 Conclusion

In this paper, we present a theoretical framework for understanding vector embedding dimensionality. We propose the PIP loss, a metric of dissimilarity between word embeddings. We focus on embedding algorithms that can be formulated as explicit or implicit matrix factorizations including the widely-used LSA, skip-gram and GloVe, and reveal a bias-variance trade-off in dimensionality selection using matrix perturbation theory. With this theory, we discover the robustness of word embeddings trained from these algorithms and its relationship to the exponent parameter $\alpha$. In addition, we propose a dimensionality selection procedure, which consists of estimating and minimizing the PIP loss. This procedure is theoretically justified, accurate and fast. All of our discoveries are concretely validated on real datasets.

**Acknoledgements** The authors would like to thank Andrea Montanari, John Duchi, Will Hamilton, Dan Jurafsky, Percy Liang, Peng Qi and Greg Valiant for the helpful discussions. We thank Balaji Prabhakar, Pin Pin Tea-mangkornpan and Feiran Wang for proofreading an earlier version and for their suggestions. Finally, we thank the anonymous reviewers for their valuable feedback and suggestions.

## Footnotes

[1]A detailed discussion on the PIP loss and analogy/relatedness is deferred to the appendix

[2]`http://www.offconvex.org/2016/02/14/word-embeddings-2/`

[3]Code can be found on GitHub: https://github.com/ziyin-dl/word-embedding-dimensionality-selection

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
