[Supplementary Material · nips_appendix.pdf]

# 7 Appendix

## 7.1 Relation between the PIP Loss and Word Analogy, Relatedness

We need to show that if the PIP loss is close to 0, *i.e.* $\|EE^T - FF^T\| \approx 0$, then $F \approx ET$ for some unitary matrix $T$. Let $E = UDV^T$ and $F = X\Lambda Y^T$ be the SVDs, we claim that we only need to show $U \approx X$ and $D \approx \Lambda$. The reason is, if we can prove the claim, then $EVY^T \approx F$, or $T = VY^T$ is the desired unitary transformation. We prove the claim by induction, assuming the singular values are simple. Note the PIP loss equals

$$\|EE^T - FF^T\| = \|UD^2U^T - X\Lambda^2 X^T\|$$

where $\Lambda = diag(\lambda_i)$ and $D = diag(d_i)$. Without loss of generality, suppose $\lambda_1 \geq d_1$. Now let $x_1$ be the first column of $X$, namely, the singular vector corresponding to the largest singular value $\lambda_1$. Regard $EE^T - FF^T$ as an operator, we have

$$\|FF^Tx_1\| - \|EE^Tx_1\| \leq \|(EE^T - FF^T)x_1\|$$
$$\leq \|EE^T - FF^T\|_{op}$$
$$\leq \|EE^T - FF^T\|_F$$

Now, notice

$$\|FF^Tx_1\| = \|X\Lambda^2 X^Tx_1\| = \lambda_1^2,$$

$$\|EE^Tx_1\| = \|UD^2U^Tx_1\| = \sum_{i=1}^n d_i^2\langle u_i, x_1\rangle \leq d_1^2 \qquad (1)$$

So $0 \leq \lambda_1^2 - d_1^2 \leq \|EE^T - FF^T\| \approx 0$. As a result, we have

1. $d_1 \approx \lambda_1$

2. $u_1 \approx x_1$, in order to achieve equality in eqn (1)

This argument can then be repeated using the Courant-Fischer minimax characterization for the rest of the singular pairs. As a result, we showed that $U \approx X$ and $D \approx \Lambda$, and hence the embedding $F$ can indeed be obtained by applying a unitary transformation on $E$, or $F \approx ET$ for some unitary $T$, which ultimately leads to the fact that analogy and relatedness are similar, as they are both invariant under unitary operations.

**Lemma 4.** For orthogonal matrices $X_0 \in \mathbb{R}^{n \times k}, Y_1 \in \mathbb{R}^{n \times (n-k)}$, the SVD of their inner product equals

$$\text{SVD}(X_0^TY_1) = U_0 \sin(\Theta)\tilde{V}_1^T$$

where $\Theta$ are the principal angles between $X_0$ and $Y_0$, the orthonormal complement of $Y_1$.

## 7.2 Proof of Lemma 4

*Proof.* We prove this lemma by obtaining the eigendecomposition of $X_0^TY_1(X_0^TY_1)^T$:

$$X_0^TY_1Y_1^TX_0 = X_0^T(I - Y_0Y_0^T)X_0$$
$$= I - U_0 \cos^2(\Theta)U_0^T$$
$$= U_0 \sin^2(\Theta)U_0^T$$

Hence the $X_0^TY_1$ has singular value decomposition of $U_0 \sin(\Theta)\tilde{V}_1^T$ for some orthogonal $\tilde{V}_1$. □

## 7.3 Proof of Lemma 1

*Proof.* Note $Y_0 = UU^TY_0 = U(\begin{bmatrix} X_0^T \\ X_1^T \end{bmatrix} Y_0)$, so

$$Y_0Y_0^T = U(\begin{bmatrix} X_0^T \\ X_1^T \end{bmatrix} Y_0Y_0^T \begin{bmatrix} X_0 & X_1 \end{bmatrix})U^T$$
$$= U \begin{bmatrix} X_0^TY_0Y_0^TX_0 & X_0^TY_0Y_0^TX_1 \\ X_1^TY_0Y_0^TX_0 & X_1^TY_0Y_0^TX_1 \end{bmatrix} U^T$$

Let $X_0^T Y_0 = U_0 \cos(\Theta) V_0^T$, $Y_0^T X_1 = V_0 \sin(\Theta) \tilde{U}_1^T$ by Lemma 4. For any unit invariant norm,

$$\left\| Y_0 Y_0^T - X_0 X_0^T \right\|$$

$$= \left\| U \left( \begin{bmatrix} X_0^T Y_0 Y_0^T X_0 & X_0^T Y_0 Y_0^T X_1 \\ X_1^T Y_0 Y_0^T X_0 & X_1^T Y_0 Y_0^T X_1 \end{bmatrix} - \begin{bmatrix} I & 0 \\ 0 & 0 \end{bmatrix} \right) U^T \right\|$$

$$= \left\| \begin{bmatrix} X_0^T Y_0 Y_0^T X_0 & X_0^T Y_0 Y_0^T X_1 \\ X_1^T Y_0 Y_0^T X_0 & X_1^T Y_0 Y_0^T X_1 \end{bmatrix} - \begin{bmatrix} I & 0 \\ 0 & 0 \end{bmatrix} \right\|$$

$$= \left\| \begin{bmatrix} U_0 \cos^2(\Theta) U_0^T & U_0 \cos(\Theta) \sin(\Theta) \tilde{U}_1^T \\ \tilde{U}_1 \cos(\Theta) \sin(\Theta) U_0^T & \tilde{U}_1 \sin^2(\Theta) \tilde{U}_1^T \end{bmatrix} - \begin{bmatrix} I & 0 \\ 0 & 0 \end{bmatrix} \right\|$$

$$= \left\| \begin{bmatrix} U_0 & 0 \\ 0 & \tilde{U}_1 \end{bmatrix} \begin{bmatrix} -\sin^2(\Theta) & \cos(\Theta)\sin(\Theta) \\ \cos(\Theta)\sin(\Theta) & \sin^2(\Theta) \end{bmatrix} \begin{bmatrix} U_0 & 0 \\ 0 & \tilde{U}_1 \end{bmatrix}^T \right\|$$

$$= \left\| \begin{bmatrix} -\sin^2(\Theta) & \cos(\Theta)\sin(\Theta) \\ \cos(\Theta)\sin(\Theta) & \sin^2(\Theta) \end{bmatrix} \right\|$$

$$= \left\| \begin{bmatrix} \sin(\Theta) & 0 \\ 0 & \sin(\Theta) \end{bmatrix} \begin{bmatrix} -\sin(\Theta) & \cos(\Theta) \\ \cos(\Theta) & \sin(\Theta) \end{bmatrix} \right\|$$

$$= \left\| \begin{bmatrix} \sin(\Theta) & 0 \\ 0 & \sin(\Theta) \end{bmatrix} \right\|$$

On the other hand by the definition of principal angles,

$$\| X_0^T Y_1 \| = \| \sin(\Theta) \|$$

So we established the lemma. Specifically, we have

1. $\| X_0 X_0^T - Y_0 Y_0^T \|_2 = \| X_0^T Y_1 \|_2$
2. $\| X_0 X_0^T - Y_0 Y_0^T \|_F = \sqrt{2} \| X_0^T Y_1 \|_F$

$\square$

Without loss of soundness, we omitted in the proof sub-blocks of identities or zeros for simplicity. Interested readers can refer to classical matrix CS-decomposition texts, for example Stewart and Sun [1990], Paige and Wei [1994], Davis and Kahan [1970], Kato [2013], for a comprehensive treatment of this topic.

### 7.4 Proof of Theorem 2

*Proof.* Let $E = X_0 D_0^\alpha$ and $\hat{E} = Y_0 \tilde{D}_0^\alpha$, where for notation simplicity we denote $D_0 = D_{1:d,1:d} = diag(\lambda_1, \cdots, \lambda_d)$ and $\tilde{D}_0 = \tilde{D}_{1:k,1:k} = diag(\tilde{\lambda}_1, \cdots, \tilde{\lambda}_k)$, with $k \leq d$. Observe $D_0$ is diagonal and the entries are in descending order. As a result, we can write $D_0$ as a telescoping sum:

$$D_0^\alpha = \sum_{i=1}^{k} (\lambda_i^\alpha - \lambda_{i+1}^\alpha) I_{i,i}$$

where $I_{i,i}$ is the $i$ by $i$ dimension identity matrix and $\lambda_{d+1} = 0$ is adopted. As a result, we can telescope the difference between the PIP matrices. Note we again split $X_0 \in \mathbb{R}^{n \times d}$ into $X_{0,0} \in \mathbb{R}^{n \times k}$ and $X_{0,1} \in \mathbb{R}^{n \times (d-k)}$, together with $D_{0,0} = diag(\lambda_1, \cdots, \lambda_k)$ and $D_{0,1} = diag(\lambda_{k+1}, \cdots, \lambda_d)$, to match the dimension of the trained embedding matrix.

$$\| E E^T - \hat{E} \hat{E}^T \|$$

$$= \| X_{0,1} D_{0,1}^{2\alpha} X_{0,1}^T - Y_0 \tilde{D}_0^{2\alpha} Y_0^T + X_{0,2} D_{0,2}^{2\alpha} X_{0,2}^T \|$$

$$\leq \| X_{0,2} D_{0,2}^{2\alpha} X_{0,2}^T \| + \| X_{0,1} D_{0,1}^{2\alpha} X_{0,1}^T - Y_0 \tilde{D}_0^{2\alpha} Y_0^T \|$$

$$= \| X_{0,2} D_{0,2}^{2\alpha} X_{0,2}^T \|$$
$$\quad + \| X_{0,1} D_{0,1}^{2\alpha} X_{0,1}^T - Y_0 D_{0,1}^{2\alpha} Y_0^T + Y_0 D_{0,1}^{2\alpha} Y_0^T - Y_0 \tilde{D}_0^{2\alpha} Y_0^T \|$$

$$\leq \| X_{0,2} D_{0,2}^{2\alpha} X_{0,2}^T \| + \| X_{0,1} D_{0,1}^{2\alpha} X_{0,1}^T - Y_0 D_{0,1}^{2\alpha} Y_0^T \|$$
$$\quad + \| Y_0 D_{0,1}^{2\alpha} Y_0^T - Y_0 \tilde{D}_0^{2\alpha} Y_0^T \|$$

We now approximate the above 3 terms separately.

1. Term 1 can be computed directly:

$$\|X_{0,2} D_{0,2}^{2\alpha} X_{0,2}^T\| = \sqrt{\|\sum_{i=k+1}^{d} \lambda_i^{2\alpha} x_{\cdot,i} x_{\cdot,i}^T\|^2} = \sqrt{\sum_{i=k+1}^{d} \lambda_i^{4\alpha}}$$

2. We bound term 2 using the telescoping observation and lemma 1:

$$\|X_{0,1} D_{0,1}^{2\alpha} X_{0,1}^T - Y_0 D_{0,1}^{2\alpha} Y_0^T\|$$

$$= \|\sum_{i=1}^{k} (\lambda_i^{2\alpha} - \lambda_{i+1}^{2\alpha})(X_{\cdot,1:i} X_{\cdot,1:i}^T - Y_{\cdot,1:i} Y_{\cdot,1:i}^T)\|$$

$$\leq \sum_{i=1}^{k} (\lambda_i^{2\alpha} - \lambda_{i+1}^{2\alpha})\|X_{\cdot,1:i} X_{\cdot,1:i}^T - Y_{\cdot,1:i} Y_{\cdot,1:i}^T\|$$

$$= \sqrt{2} \sum_{i=1}^{k} (\lambda_i^{2\alpha} - \lambda_{i+1}^{2\alpha})\|Y_{\cdot,1:i}^T X_{\cdot,i:n}\|$$

3. Third term:

$$\|Y_0 D_{0,1}^{2\alpha} Y_0^T - Y_0 \tilde{D}_0^{2\alpha} Y_0^T\| = \sqrt{\|\sum_{i=1}^{k} (\lambda_i^{2\alpha} - \tilde{\lambda}_i^{2\alpha}) Y_{\cdot,i} Y_{\cdot,i}^T\|^2}$$

$$= \sqrt{\sum_{i=1}^{k} (\lambda_i^{2\alpha} - \tilde{\lambda}_i^{2\alpha})^2}$$

Collect all the terms above, we arrive at an approximation for the PIP discrepancy:

$$\|EE^T - \hat{E}\hat{E}^T\| \leq \sqrt{\sum_{i=k+1}^{d} \lambda_i^{4\alpha}} + \sqrt{\sum_{i=1}^{k} (\lambda_i^{2\alpha} - \tilde{\lambda}_i^{2\alpha})^2}$$

$$+ \sqrt{2} \sum_{i=1}^{k} (\lambda_i^{2\alpha} - \lambda_{i+1}^{2\alpha})\|Y_{\cdot,1:i}^T X_{\cdot,i:n}\|$$

$\square$

## 7.5 Proof of Lemma 2

*Proof.* To bound the term $\sqrt{\sum_{i=1}^{k} (\lambda_i^{2\alpha} - \tilde{\lambda}_i^{2\alpha})^2}$, we use a classical result [Weyl, 1912, Mirsky, 1960].

**Theorem 4** (Weyl)**.** Let $\{\lambda_i\}_{i=1}^{n}$ and $\{\tilde{\lambda}_i\}_{i=1}^{n}$ be the spectrum of $M$ and $\tilde{M} = M + Z$, where we include 0 as part of the spectrum. Then

$$\max_i |\lambda_i - \tilde{\lambda}_i| \leq \|Z\|_2$$

**Theorem 5** (Mirsky-Wielandt-Hoffman)**.** Let $\{\lambda_i\}_{i=1}^{n}$ and $\{\tilde{\lambda}_i\}_{i=1}^{n}$ be the spectrum of $M$ and $\tilde{M} = M + Z$. Then

$$(\sum_{i=1}^{n} |\lambda_i - \tilde{\lambda}_i|^p)^{1/p} \leq \|Z\|_{S_p}$$

We use a first-order Taylor expansion followed by applying Weyl's theorem 4:

$$\sqrt{\sum_{i=1}^{k} (\lambda_i^{2\alpha} - \tilde{\lambda}_i^{2\alpha})^2} \approx \sqrt{\sum_{i=1}^{k} (2\alpha \lambda_i^{2\alpha-1}(\lambda_i - \tilde{\lambda}_i))^2}$$

$$= 2\alpha \sqrt{\sum_{i=1}^{k} \lambda_i^{4\alpha-2}(\lambda_i - \tilde{\lambda}_i)^2}$$

$$\leq 2\alpha \|N\|_2 \sqrt{\sum_{i=1}^{k} \lambda_i^{4\alpha-2}}$$

Now take expectation on both sides and use Tracy-Widom Law:

$$\mathbb{E}[\sqrt{\sum_{i=1}^{k}(\lambda_i^{2\alpha} - \tilde{\lambda}_i^{2\alpha})^2}] \leq 2\sqrt{2n}\alpha\sigma\sqrt{\sum_{i=1}^{k}\lambda_i^{4\alpha-2}}$$

$\square$

A further comment is that this bound can tightened for $\alpha = 0.5$, by using Mirsky-Wieland-Hoffman's theorem instead of Weyl's theorem [Stewart and Sun, 1990]. In this case,

$$\mathbb{E}[\sqrt{\sum_{i=0}^{k}(\lambda_i^{2\alpha} - \tilde{\lambda}_i^{2\alpha})^2}] \leq k\sigma$$

where we can further save a $\sqrt{2n/k}$ factor.

## 7.6 Proof of Lemma 3

Classical matrix perturbation theory focuses on bounds; namely, the theory provides upper bounds on how much an invariant subspace of a matrix $\tilde{A} = A + E$ will differ from that of $A$. Note we switched notation to accommodate matrix perturbation theory conventions (where usually $A$ denotes the unperturbed matrix , $\tilde{A}$ is the one after perturbation, and $E$ denotes the noise). The most famous and widely-used ones are the $\sin\Theta$ theorems:

**Theorem 6** (sine $\Theta$). For two matrices $A$ and $\tilde{A} = A + E$, denote their singular value decompositions as $A = XDU^T$ and $\tilde{A} = Y\Lambda V^T$. Formally construct the column blocks $X = [X_0, X_1]$ and $Y = [Y_0, Y_1]$ where both $X_0$ and $Y_0 \in \mathbb{R}^{n \times k}$, if the spectrum of $D_0$ and $D_1$ has separation

$$\delta_k \overset{\Delta}{=} \min_{1 \leq i \leq k, 1 \leq j \leq n-k}\{(D_0)_{ii} - (D_1)_{jj}\},$$

then

$$\|Y_1^T X_0\| \leq \frac{\|Y_1^T E X_0\|}{\delta_k} \leq \frac{\|E\|}{\delta_k}$$

Theoretically, the sine $\Theta$ theorem should provide an upper bound on the invariant subspace discrepancies caused by the perturbation. However, we found the bounds become extremely loose, making it barely usable for real data. Specifically, when the separation $\delta_k$ becomes small, the bound can be quite large. So what was going on and how should we fix it?

In the *minimax* sense, the gap $\delta_k$ indeed dictates the max possible discrepancy, and is tight. However, the noise $E$ in our application is random, not adversarial. So the universal guarantee by the sine $\Theta$ theorem is too conservative. Our approach uses a technique first discovered by Stewart in a series of papers [Stewart and Sun, 1990, Stewart, 1990]. Instead of looking for a universal upper bound, we derive a *first order approximation* of the perturbation.

### 7.6.1 First Order Approximation of $\|Y_1^T X_0\|$

We split the signal $A$ and noise $E$ matrices into block form, with $A_{11}, E_{11} \in \mathbb{R}^{k \times k}$, $A_{12}, E_{12} \in \mathbb{R}^{k \times (n-k)}$, $A_{21}, E_{21} \in \mathbb{R}^{(n-k) \times k}$ and $A_{22}, E_{22} \in \mathbb{R}^{(n-k) \times (n-k)}$.

$$A = \left[\begin{array}{c|c} A_{11} & A_{12} \\ \hline A_{21} & A_{22} \end{array}\right], \quad E = \left[\begin{array}{c|c} E_{11} & E_{12} \\ \hline E_{21} & E_{22} \end{array}\right]$$

As noted by Stewart in [Stewart, 1990],

$$X_0 = Y_0(I + P^T P)^{\frac{1}{2}} - X_1 P \tag{2}$$

and

$$Y_1 = (X_1 - X_0 P^T)(I + P^T P)^{-\frac{1}{2}} \tag{3}$$

where $P$ is the solution to the equation

$$T(P) + (E_{22}P - PE_{11}) = E_{21} - P\tilde{A}_{12}P \tag{4}$$

The operator $T$ is a linear operator on $P \in \mathbb{R}^{(n-k) \times k} \to \mathbb{R}^{(n-k) \times k}$, defined as

$$T(P) = A_{22}P - PA_{11}$$

Now, we drop the second order terms in equation (2) and (3),

$$X_0 \approx Y_0 - X_1 P, \ Y_1 \approx X_1 - X_0 P^T$$

So

$$Y_1^T X_0 \approx Y_1^T (Y_0 - X_1 P) = Y_1^T X_1 P$$
$$\approx (X_1^T - P X_0^T) X_1 P = P$$

As a result, $\|Y_1^T X_0\| \approx \|P\|$.
To approximate $P$, we drop the second order terms on $P$ in equation (4), and get:

$$T(P) \approx E_{21} \tag{5}$$

or $P \approx T^{-1}(E_{21})$ as long as $T$ is invertible. Our final approximation is

$$\|Y_1^T X_0\| \approx \|T^{-1}(E_{21})\| \tag{6}$$

### 7.6.2 The Sylvester Operator $T$

To solve equation (6), we perform a spectral analysis on $T$:

**Lemma 5.** There are $k(n-k)$ eigenvalues of $T$, which are

$$\lambda_{ij} = (D_0)_{ii} - (D_1)_{jj}$$

*Proof.* By definition, $T(P) = \lambda P$ implies

$$A_{22} P - P A_{11} = \lambda P$$

Let $A_{11} = U_0 D_0 U_0^T$, $A_{22} = U_1 D_1 U_1^T$ and $\tilde{P} = U_1^T P U_0$, we have

$$D_0 \tilde{P} - \tilde{P} D_1 = \lambda \tilde{P}$$

Note that when $\tilde{P} = e_i e_j^T$,

$$D_0 e_i e_j^T - e_i e_j^T D_1 = ((D_0)_{ii} - (D_1)_{jj}) e_i e_j^T$$

So we know that the operator $T$ has eigenvalue $\lambda_{ij} = (D_0)_{ii} - (D_1)_{jj}$ with eigen-function $U_1 e_i e_j^T U_0^T$. $\qquad \square$

Lemma 5 not only gives an orthogonal decomposition of the operator $T$, but also points out when $T$ is invertible, namely the spectrum $D_0$ and $D_1$ do not overlap, or equivalently $\delta_k > 0$. Since $E_{12}$ has iid entries with variance $\sigma^2$, using lemma 5 together with equation (6) from last section, we conclude

$$\|Y_1^T X_0\| \approx \|T^{-1}(E_{21})\|$$
$$= \| \sum_{1 \le i \le k, 1 \le j \le n-k} \lambda_{ij}^{-1} \langle E_{21}, e_i e_j^T \rangle \|$$
$$= \sqrt{ \sum_{1 \le i \le k, 1 \le j \le n-k} \lambda_{ij}^{-2} E_{21,ij}^2 }$$

By Jensen's inequality,

$$\mathbb{E}\|Y_1^T X_0\| \le \sqrt{\sum_{i,j} \lambda_{ij}^{-2} \sigma^2} = \sigma \sqrt{\sum_{i,j} \lambda_{ij}^{-2}}$$

Our new bound is *much sharper* than the sine $\Theta$ theorem, which gives $\sigma \frac{\sqrt{k(n-k)}}{\delta}$ in this case. Notice if we upper bound every $\lambda_{ij}^{-2}$ with $\delta_k^{-2}$ in our result, we will obtain the same bound as the sine $\Theta$ theorem. In other words, our bound considers *every* singular value gap, not only the smallest one. This technical advantage can clearly be seen, both in the simulation and in the real data.

## 7.7 Growth Rate Analysis of the Variance Terms

The second term $2\sqrt{2n}\alpha\sigma\sqrt{\sum_{i=1}^{k}\lambda_i^{4\alpha-2}}$ increases with respect to $k$ at rate of $\lambda_k^{2\alpha-1}$. Not as obvious as the second term, the last term also increases at the same rate. Note in $(\lambda_k^{2\alpha} - \lambda_{k+1}^{2\alpha})\sqrt{\sum_{r\leq k<s}(\lambda_r - \lambda_s)^{-2}}$, the square root term is dominated by $(\lambda_k - \lambda_{k+1})^{-1}$ which gets closer to infinity as $k$ gets larger. On the other hand, $\lambda_k^{2\alpha} - \lambda_{k+1}^{2\alpha}$ can potentially offset this first order effect. Specifically, consider the smallest non-zero singular value $\lambda_d$, whose gap to 0 is $\lambda_d$. Note when the two terms are multiplied,

$$(\lambda_d^{2\alpha} - 0)(\lambda_d - 0)^{-1} = \lambda_d^{2\alpha-1},$$

which shows the two variance terms have the same rate of $\lambda_k^{2\alpha-1}$.

## 7.8 Experimentation Setting for Dimensionality Selection Time Comparison

For PIP loss minimizing method, we first estimate the spectrum of $M$ and noise standard deviation $\sigma$ with methods described in Section 5.2.1. $E = UD^\alpha$ was generated with a random orthogonal matrix $U$. Note any orthogonal $U$ is equivalent due to the unitary invariance. For every dimensionality $k$, the PIP loss for $\hat{E} = \tilde{U}_{\cdot,1:k}\tilde{D}_{1:k,1:k}^\alpha$ was calculated and $\|\hat{E}\hat{E}^T - EE^T\|$ is computed. Sweeping through all $k$ is very efficient because one pass of full sweeping is equivalent of doing a single SVD on $\tilde{M} = M + Z$. The method is the same for LSA, skip-gram and GloVe, with different signal matrices (PPMI, PMI and log-count respectively).

For empirical selection method, the following approaches are taken:

- LSA: The PPMI matrix is constructed from the corpus, a full SVD is done. We truncate the SVD at $k$ to get dimensionality $k$ embedding. This embedding is then evaluated on the testsets [Halawi et al., 2012, Finkelstein et al., 2001], and each testset will report an optimal dimensionality. Note the different testsets may not agree on the same dimensionality.

- Skip-gram and GloVe: We obtained the source code from the authors' Github repositories[4][5]. We then train word embeddings from dimensionality 1 to 400, at an increment of 2. To make sure all CPUs are effectively used, we train multiple models at the same time. Each dimensionality is trained for 15 epochs. After finish training all dimensionalities, the models are evaluated on the testsets [Halawi et al., 2012, Finkelstein et al., 2001, Mikolov et al., 2013a], where each testset will report an optimal dimensionality. Note we already used a step size larger than 1 (2 in this case) for dimensionality increment. Had we used 1 (meaning we train every dimensionality between 1 and 400), the time spent will be doubled, which will be close to a week.

## Footnotes

[4]https://github.com/tensorflow/models/tree/master/tutorials/embedding

[5]https://github.com/stanfordnlp/GloVe