[Reviews · NeurIPS 2018]

Reviewer 1



This is indeed one of the best papers I've read in a while. The paper simply studies how the dimensionality of word-embeddings relates to the bias-variance trade-off. The paper builds upon the fact that word-embeddings are unitary invariant. Thus, the technique to compare two word embeddings should also respect this property. As a result, the paper introduces the PIP loss metric which simply computes the cosine similarity matrices of words for both embeddings, subtracts the two matrix and compute the norm. The significant contributions are that, based on this loss metric, they can describe the bias-variance trade-off when selecting the number of dimensions. Moreover, by minimizing the PIP loss, one can determine the ideal dimensionality for their setting. The paper is well-presented and the results are strong, and I think the NIPS community would benefit from reading the paper.

Reviewer 2



In this work a Pairwise Inner Product Loss is developed, motivated by unitary invariance of word embeddings. It then investigates theoretically the relationship between word embedding dimensionality to robustness under different singular exponents, and relates it to bias/variance tradeoff. The discovered relationships are used to define a criterion for a word embedding dimensionality selection procedure, which is empirically validated on 3 intrinsic evaluation tasks. The PIP loss technique is well motivated, clear, and easy to understand. It would be interesting to see this technique applied in other contexts and for other NLP tasks. The paper is clearly written, well motivated, and sections follow naturally. Only section 5.2 would benefit from some more details (which are deferred to the appendix, presumably due to space constraints), and the Figures are very small. To improve the latter point, I recommend increasing font sizes of axis labels and ticks, as well as legend, and line width. The work is original and provides a new theoretical view on the robustness of word embeddings. Perhaps the title "On Word Embedding Dimensionality" is a bit too broad as the work chooses a definite route, even though it reveals connections between several existing approaches. A potential criticism of this work is that it empirically only investigates word embedding dimensionality using intrinsic evaluation (though on 3 tasks), but not extrinsically / in a downstream NLP task. This is even though understanding the effect of word embedding dimensionality on downstream models and applications is one of the major motivations laid out in the Introduction. Second, it is acknowledged that empirical validation carries "the risk of selecting sub-optimal dimensionalities" (line 37), but a similar caveat is missing for the proposed method "We can explicitly answer the open question of dimensionality selection for word embedding" (line 10-11). It should be kept in mind that also the proposed method (even though theoretically motivated) also only minimises an _expected_ quantity, and that it is also prone to the same risk of choosing sub-optimal dimensionality as a purely empirical selection. Related to the previous point - another potential criticism is that the PIP Loss metric does not take into account word frequency. Less frequent words will have embeddings with more variance than frequent words. But by Zipf's Law, most words are rare, and giving all words the same contribution in the PIP loss will result in increased variance of the PIP loss when word embeddings are fitted on different corpora. This is speculative, but perhaps it is worth down-weighting infrequent word dimensions in the PIP loss? To investigate this possible phenomenon one could fit vectors either other datasets than text8, or different portions of text8, and observe confidence intervals. Minor comments: - Line 227: could you explain why the square root term is dominated by (\lambda_k - \lambda_{k-1})^{-1} - Could you expand on line 166 (third line)? - Line 227 (end) -- should the exponent be -2? - to avoid misinterpretation, one could be more explicit in which exact term is plotted in Figure 1a. - line 254: the relative gap to the optimal dimension would be more informative - line 250: Are the chosen dimensionalities really exactly the optimal ones also empirically? It would be better if this was more concrete: dimensionalities chosen with PIP loss minimisation are generally in line with empirical validation, but not exactly identical? - line 267: could you elaborate on this?

Reviewer 3



This paper conducts an in-depth analysis of the optimal dimension for word embeddings. It starts with proposing a loss, based on a very straightforward motivation, and then decomposes the loss into variance and bias components in a very nice way (which is Theorem 1 in the paper). Then the theorem gets extended to other cases, without changing the main motivation behind it, and reveals its fundamental relation to SNR in information theory and signal processing. Thus it gives a satisfying good theoretical inspection and explanation to [Arora, 2016]. Plus the support of previous research [Levy and Goldberg, 2014] on the connection between skip-gram model and matrix factorization, the whole set of theory becomes applicable to a set of widely used models like skip-gram. I would say reading through the main part of the paper is enlightening and rather an enjoyable adventure. The experimental results regarding the robustness of word2vec/glove models w.r.t. dimensions (Section 5.1) are consistent with what many people are experiencing when using these word embeddings. Which the paper has well explained that in its previous sections. I have to say it is a very good presentation of an in-depth explanation of the well-known word embedding tasks. And the theoretical results do have its practical applications.